

# A representative density profile for the North Greenland snowpack

Christoph Florian Schaller[1], Johannes Freitag[1], Sepp Kipfstuhl[1], Thomas Laepple[2], Hans Christian Steen-Larsen[3], and Olaf Eisen[1,4]

[1]Alfred Wegener Institute, Helmholtz Centre for Polar and Marine Research, Bremerhaven, Germany
[2]Alfred Wegener Institute, Helmholtz Centre for Polar and Marine Research, Potsdam, Germany
[3]Centre for Ice and Climate, Niels Bohr Institute, University of Copenhagen, Denmark
[4]Department of Geosciences, University of Bremen, Germany

*Correspondence to:* Christoph Schaller (christoph.schaller@awi.de)

**Abstract.** Along a traverse through North Greenland in May 2015 we sampled the top two meters of snow and analyzed its density and water isotopic composition. A new technique for probing the upper meters of the snow and an adapted algorithm for comparing data sets from different positions and aligning stratigraphic features is presented. We find good agreement of the density layering in the snowpack over hundreds of kilometers, which allows the construction of a representative density profile.

The results are supported by an empirical based statistical density model, that is used to generate sets of random profiles and validate the applied methods. Furthermore we are able to calculate annual accumulation rates, align melt layers and observe isotopic temperatures in the area back to 2010. Distinct relations of $\delta^{18}O$ with both accumulation rate and density are deduced. Inter alia the depths of the 2012 melt layers and high resolution densities are provided for applications in remote sensing.

## 1 Introduction

In the context of global warming, the Greenland ice sheet has been identified as a so called "tipping point" of climate change (Lenton et al., 2008). The sea level rise caused by its decay may have severe impact on human society as well as ecological systems. Thus the difference in accumulation across the interior of the ice sheet and seasonal melting, runoff and calving at its borders, the so called mass balance, has been in the focus of recent scientific activities in the Arctic region. The applied methods for its determination range from satellite remote sensing (e.g. Zwally et al., 2011), over regional climate modeling

(e.g. Fettweis, 2007) to large scale climate simulations constrained by weather station data and ice core records (e.g. Hanna et al., 2011). Even though first accumulation and density measurements were already carried out in 1952 – 54 (Bull, 1958) using accumulation stakes and Rammsonde measurements at a few points alongside the gravity survey of the British North Greenland Expedition, large scale studies such as Benson (1962) are still very rare. To obtain accumulation maps of Greenland such as Bales et al. (2009) diverse data sets from ice cores, snow pits and weather stations have to be collected over several

20 years. Recently Hawley et al. (2014) conducted a ground-penetrating radar survey alongside a traverse of about 1000 km length, supported by a few snow pits and shallow cores for bulk densities and chemical profiling. Koenig et al. (2015) used airborne snow radar to determine accumulation rates from 2009 to 2012 along flight paths of more than ten thousand kilometers.





In summer 2012, there were two very warm days with temperatures above 0°C almost all over Greenland, causing substantial melt layers (Nghiem et al., 2012). Although this was a very rare event induced by a special weather situation (Bennartz et al., 2013), the newly formed ice layers strongly influence the physical properties of the snow and firn pack and thereby also measurements (Nilsson et al., 2015).

We introduce a new and efficient technique for sampling the snowpack along traverses, which allows for additional lab-based measurements to gain high resolution profiles of physical snow properties, such as density. Furthermore we adapt an algorithm from speech recognition to align those spatially distributed data sets and provide further insight into their development with changing surrounding conditions. The method is tested with randomly generated sets of density profiles with the same statistical properties as the original measurements. As an application we present data gained along a 450 km traverse in North Greenland,

deduce relations of the individual parameters and show additional values of interest such as the depths of the 2012 melt layers.

## 2 Data acquisition and processing

In preparation for the upcoming East GReenland Ice core Project (EGRIP), the Danish Center for Ice and Climate's dome and equipment had to be moved about 450 km from the previous drilling site, NEEM. Alongside this so called "N2E" traverse in May 2015 several measurements of the upper part of the firn and the snow surface were undertaken. Amongst others, the

15 upper two meters of the snowpack were sampled using the "liner technique" described in detail below. Snow cores were taken approximately every 25 km at the positions shown in Fig. 1, detailed coordinates can be found in Table 1.

### 2.1 Liner technique

The sampling is done using carbon fibre tubes with sharp edges of one meter length, ten centimeters diameter and one millimeter wall thickness (called "liners"). To start off, the first liner is carefully pushed and hammered into the ground until its top

is parallel to the snow surface. Nonetheless in a few cases the snow inside the tube might be slighty compacted by up to two centimeters in the vertical direction. Subsequently a snow pit of one meter depth is dug next to the tube and the snow cut off at its bottom using a metal plate or small saw. The tube is removed and its openings sealed using matching plastic bags. Then the cutting surface is cleaned and the second liner inserted right below the first one. Finally the pit has to be deepened to two meters to once again cut off the snow and take the second liner. Theoretically the described process can be iterated up to an

arbitrary depth. However, the area of the required snow pit increases significantly with every meter of depth gained. Probing the upper two meters by that technique takes approximately two hours.

### 2.2 X-ray tomography

The cores were transported to the Alfred-Wegener-Institute, Bremerhaven, in frozen condition. All samples were analyzed in the AWI-Ice-CT (Freitag et al., 2013), a worldwide unique X-ray computer tomograph in a cold lab, which allows $\mu$m resolution

density measurements of whole one meter core segments in 2D and 3D. As part of the measurement procedure a sample holder for liners was constructed, that itself contains several pieces of pure ice of known geometry for calibration purposes. Amongst





others, the effect of the carbon fibre tube being part of the scan was corrected for, using empty tube measurements. Thus, the fragile snow cores do not have to be removed from the liners.

As the required measurement time increases with accuracy, we chose to do 2D scans with a resolution of approximately $0.128\,\mathrm{mm}$. Each of these scans takes about three minutes. However, fifteen minutes per meter are more realistic when including sample preparation and accurate documentation. Then, the raw measurement data are automatically processed by detecting the calibration unit and directly calculating densities from the the CT images. Additionally, for each liner, the mean density is determined from the weight and geometry of the snow as an independent comparison value. Figure 2 displays an example CT image with a zoomed section showing two melt layers in the snowpack aligned with the respective densities derived from 2D analysis.

## 2.3 Isotope measurements

Finally, the snow was gently pushed out of the tubes and cut in samples with a vertical height of one centimeter for the $30\,\mathrm{cm}$ right below the surface and two centimeters otherwise. These samples were crushed and sealed in plastic bags. Finally water isotopes were measured using a Picarro L2130-i with a precision of $\sigma = 0.1\,\text{‰}$ for $\delta^{18}\mathrm{O}$.

The snow was dated by determining and counting the maxima (summer) and minima (winter) in the seasonal $\delta^{18}\mathrm{O}$ signal. Using the density data, accumulation rates were calculated from the ice mass at the different sites for the contained three to five years. In the present study, we only use winter to winter rates – summer to summer values were computed as a reference but show no different behaviour.

## 3 Mathematical methods

### 3.1 Automatic alignment of stratigraphic features

In order to efficiently analyze the data sets generated along the traverse, we investigated several ways to automatically detect coherent signals at the different positions. A renowned matching method is maximizing the cross correlation. However, determining a constant shift between two profiles is not suitable for our case as the accumulation rate, and thus the vertical spacing of layers, is subject to change going eastwards. Under the assumption of constant accumulation over time and no significant compaction in the top two meters, one would expect a shift which is linearly increasing with depth and has a slope equal to the ratio of accumulation rates. Then again, local environmental conditions such as wind speed and direction influence the mass accumulated by a certain event (Fisher et al., 1985). Therefore we aimed to align snow and its properties with continuously changing shifts.

The Dynamic Time Warping (DTW) method, that was introduced to speech recognition in the seventies (Itakura, 1975), provides an efficient algorithm for that purpose. The basic idea is to discretize the two data sets to be compared with the same step size $l$ (resulting in two vectors $\mathbf{S}$ and $\mathbf{T}$ of length $n$ and $m$) and then consecutively assign the values of one to another, whereby each value can be matched with multiple values of the other data set. To find the best fit, one calculates a matrix $\underline{\mathbf{D}}$




where $\underline{\mathbf{D}}[i,j]$ indicates the error of the best path that leads to the $i$'th element of the first data set being connected to the $j$'th element of the second one.

The original algorithm starts by calculating the matrix in the upper left corner, fixing the first elements of both data sets to be linked with each other. Then it proceeds through the matrix by taking the path with the minimal error leading to the respective cell and adding the local error, i.e.

$$\underline{\mathbf{D}}[i,j] = \begin{cases} \infty & \text{for } i < 0 \text{ or } j < 0 \\ \|\mathbf{S}[0] - \mathbf{T}[0]\| & \text{for } i = 0 \text{ and } j = 0 \\ \|\mathbf{S}[i] - \mathbf{T}[j]\| + \min(\underline{\mathbf{D}}[i,j-1], \underline{\mathbf{D}}[i-1,j-1], \underline{\mathbf{D}}[i-1,j]) & \text{else.} \end{cases} \tag{1}$$

Finally, it goes to cell $\underline{\mathbf{D}}[n,m]$ and backtraces the path of minimal errors to $\underline{\mathbf{D}}[0,0]$, obtaining the best fit of the complete data sets in the given norm $\|\cdot\|$.

For our application – matching measurements of the upper two meters of the snowpack – we do not aim to fit complete data sets, but rather allow for different offsets at the top and bottom. The former may be caused by variations of the snow surface due to current conditions, the latter by different accumulation rates leading to data at the bottom of the liners not having any physical relation apart from being the deepest snow analyzed at the given position. To accomplish that, we expand the idea of Sakurai et al. (2007) introducing maximal surface and bottom index offsets $s$ and $b$. Then we initialize $\underline{\mathbf{D}}$ by

$$\underline{\mathbf{D}}[0,j] = \|\mathbf{S}[0] - \mathbf{T}[j]\| \text{ for } 0 \leq j \leq s \text{ and} \tag{2}$$

$$\underline{\mathbf{D}}[i,0] = \|\mathbf{S}[i] - \mathbf{T}[0]\| \text{ for } 0 < i \leq s \tag{3}$$

before proceeding through the matrix. Finally instead of backtracing simply from $\underline{\mathbf{D}}[n,m]$, we end our fitting path at

$$\min\{\underline{\mathbf{D}}[i,j] \,|\, (i = n \text{ and } m - b \leq j \leq m) \text{ or } (j = m \text{ and } n - b \leq i \leq n)\} \tag{4}$$

and search a trace back to any of the initialized elements. Thereby we find the best matching of subsets of $S$ and $T$ with a maximal shift of $s \cdot l$ at the top and $b \cdot l$ at the bottom. In between, we verify that a linearly increasing maximal shift is not exceeded.

The simple way we proceed through the matrix so far, often refered to as "stepping pattern", is unrealistic for our case as a single value of one data set could be fit to arbitrary many values of the other data set. Along the traverse we find the maximal ratio of the respective accumulation rates to be a little smaller than two. Therefore, we apply a constrained stepping as presented by Sakoe and Chiba (1978) such that each value of one data set can be fit to at most two values of the other. This is obtained by

$$\underline{\mathbf{D}}[i,j] = \begin{cases} \|\mathbf{S}[i] - \mathbf{T}[j]\| + \min(\underline{\mathbf{D}}[i,j-1], \underline{\mathbf{D}}[i-1,j-1], \underline{\mathbf{D}}[i-1,j]) & \text{for } i = 1 \text{ or } j = 1 \\ \|\mathbf{S}[i] - \mathbf{T}[j]\| + \min \begin{pmatrix} \|\mathbf{S}[i-1] - \mathbf{T}[j]\| + \underline{\mathbf{D}}[i-2,j-1] \\ \underline{\mathbf{D}}[i-1,j-1] \\ \|\mathbf{S}[i] - \mathbf{T}[j-1]\| + \underline{\mathbf{D}}[i-1,j-2] \end{pmatrix} & \text{else.} \end{cases} \tag{5}$$





Figure 3 illustrates the different patterns for proceeding through the matrix. In the aftermath, the backtracing has to occur according to the implemented stepping.

Finally, we do not only want to fit one type of data (e.g. densities) but combine all the available information in the profiles to gain a robust picture of the developing stratigraphy along the traverse. In a first step, we match the $\delta^{18}$O signal, which shows a clear seasonal behavior but almost no small scale variations as the high frequency component is lost by diffusion. Then, we use the obtained depth assignment of the two different positions to resample the measured densities to a common depth scale. In a second step, we apply the algorithm to these densities at a much higher resolution to fine tune our depth alignment according to small scale stratigraphic features. As a norm we use the Euclidean distance divided by the path length (i.e the root mean square error), which means that we have to keep track of the path lengths in a second matrix. Table 2 summarizes the final set of parameters.

This method does not only allow us to compare data from two positions, but also to obtain a moving depth scale by fitting the liners to the first data set one by one. The result, a continuous image of the snow layering, can be compared with other indicators such as the melt layer positions. In addition, being able to align densities and stratigraphic features all along the traverse enables us to provide a representative density profile for the region. For its construction, we first use the continuous layering to transform all density curves to the first depth scale (NEEM) and average them. This, however, is not yet a representative density profile as all profiles now replicate the layering at NEEM, e.g. a layer that is very thin there but thicker at most positions would be considered thin. To overcome this, we calculate the mean shifts applied to the values that were aligned and thus averaged. On average, i.e. for constant accumulation rates, we would expect these shifts to go linear with depth for the layering to be representative. Thus we calculate a linear least squares regression and correct the depth scale accordingly.

Nonetheless, the depth scale still represents the accumulation rate at NEEM. To transfer the average profile to location $X$, we need to calculate a linear rescaling factor $f_X$ for the depth $d_X$ that fulfills

$$d_{NEEM} = d_X \cdot f_X. \tag{6}$$

We expect $f_X$ to be determined by the accumulation rate, or rather its ratio to the one at NEEM.

## 3.2 Significance testing and surrogate density profiles

Any alignment method will increase the covariance between records even if they are not related (Haam and Huybers, 2010). Therefore, to test the statistical significance of our density alignment, we generate sets of surrogate density profiles with similar statistical properties for each position and process them the same way as the real data.

The complexity of the density signal consisting of slow variations, sharp layer changes as well as strong melt layer and wind crust related density spikes inhibits the use of simple surrogate construction methods such as autoregressive processes. Instead we propose the following algorithm.

As a base curve, we identify the $\delta^{18}$O component of the density signal by linear regression, using the same step size $l_{low}$ as for the coarse ($\delta^{18}$O based) fitting step. This can be done because we rely on $\delta^{18}$O to follow a seasonal cycle – otherwise water isotope dating would be impossible. Let $\rho_{base}$ be the base density from $\delta^{18}$O, $r_{low}$ the autocorrelation and $\sigma_{base}$ the standard





deviation of the fluctuations of the measured density (averaged to resolution $l_{low}$) around $\rho_{base}$ for lag $l_{low}$. We start generating an artificial low resolution density profile $\rho_{low}$ by

$$\rho_{low}(x_i) = \rho_{base}(x_i) + \varepsilon_i \tag{7}$$

$$\varepsilon_i = \begin{cases} \nu_0 & \text{for } i = 0 \\ r_{low} \cdot \varepsilon_{i-1} + \nu_i & \text{else} \end{cases} \tag{8}$$

$$\nu \sim \mathcal{N}(0, \sigma_{base}) \tag{9}$$

where $x_i = x_0 + i \cdot l_{low}$. $\tag{10}$

Here $\nu \sim \mathcal{N}(0, \sigma_{base})$ implies that the $\nu_i$ are distributed normally with mean 0 and standard deviation $\sigma_{base}$. In the following, $\mathcal{U}(0,1)$ will represent a continuous uniform distrubtion for the interval $[0,1]$. The inclusion of higher autocorrelation lengths is straigthforward. $r_{low}$ has to be replaced by the autocorrelation matrix, which is multiplied with a vector of the preceding $\varepsilon_i$. Second, on the fine scale (step size $l_{high}$), we have a look at the differences between the interpolated low resolution density and the high resolution density values in the measurements. As we find the distribution to be trimodal, we split the differences in three components - low amplitude variations within the same layer (henceforth denoted "noise" even though they might partly have physical origin), fast and moderate amplitude changes in the density at layer transitions or wind crusts ("shocks") and rapid high amplitude changes at melt layers ("melt"). Again, we compute the autocorrelation factor $r_{high}$ for lag $l_{high}$. Nonetheless, this time, the standard deviations $\sigma_{noise}$, $\sigma_{shocks}$ and $\sigma_{melt}$ and the means $\mu_{shocks}$ and $\mu_{melt}$ have to be calculated separately. Furthermore we need to estimate the probabilities $P_{shocks}$ and $P_{melt}$ of beginning a shock or a melt layer at a specific position. For this purpose, we determine the number of melt layers $N_{melt}$, the number of shocks $N_{shocks}$ and the average distance to the previous shock $d_{avg}$. In addition, we denote the total number of data points by $N$ and the distance to the last shock at a given position $i$ by $d_i$. Finally, the basic model to generate a random density profile $\rho_{high}$ is

$$\rho_{high}(x_i) = \rho_{low}(x_i) + \kappa_i \tag{11}$$

$$\kappa_i = \begin{cases} \phi_i & \text{for } i = 0 \text{ or } P > P_{melt} + P_{shocks} \\ \mathcal{N}(\mu_{shocks}, \sigma_{shocks}) & \text{for } i \neq 0 \text{ and } P_{melt} < P \leq P_{melt} + P_{shocks} \\ \mathcal{N}(\mu_{melt}, \sigma_{melt}) & \text{for } i \neq 0 \text{ and } P \leq P_{melt} \end{cases} \tag{12}$$

$$P_{melt} = \frac{N_{melt}}{N} \tag{13}$$

$$P_{shocks} = \frac{d_i}{d_{avg}} \cdot \frac{N_{shocks}}{N} \tag{14}$$

$$P \sim \mathcal{U}(0,1) \tag{15}$$

$$\phi_i = \begin{cases} \nu_0 & \text{for } i = 0 \\ r_{high} \cdot \phi_{i-1} + \nu_i & \text{else} \end{cases} \tag{16}$$

$$\nu \sim \mathcal{N}(0, \sigma_{base}) \tag{17}$$





where $x_i = x_0 + i \cdot l_{high}$. (18)

The same approach as before can be used to expand to higher autocorrelation lengths. However, we use the model in the presented form as it already provides realistic density surrogates.

# 4 Results

## 4.1 Profile alignment

As an example of the matching process, we present a fit of data from N2E_11 to the first position (NEEM) in Fig. 4. The distance between the two locations is approximately $240\,\mathrm{km}$, i.e. a little more than half of the total traverse length. First the $\delta^{18}$O profiles are matched, yielding an approximately linearly increasing coarse shift. In the second step the densities are fine tuned, which results in small shifts fluctuating around zero and never reaching the allowed maximum of $0.1\,\mathrm{m}$. To provide an overview of the changing snow structure, we fit all combinations of two liners and plotted the matrix of the root mean square errors (RMSE) of in Fig. 5. A noticeable change in the pattern of the fitting errors occurs between the fourth and fifth position along the traverse.

Figure 6 shows the continuous depth scale obtained by fitting all liners along the traverse to the first position (NEEM). For comparison, the melt layer positions detected during the CT measurements (cf. Table 3) have been included. In addition, selected density profiles are displayed. Using the previously calculated depth scale density records were stacked to obtain a representative density profile (Fig. 7). The gray area indicates a one standard deviation error band. Comparing the necessary rescaling factors (known from the construction of the stacked profile) to the ratio of accumulation rates, we apply linear least squares to find

$$f_X = 0.325 + 0.665 \cdot \frac{\dot{a}_{NEEM}}{\dot{a}_X}$$
(19)

where $\dot{a}_X$ denotes the mean annual accumulation rate at position $X$. The coefficient of determination is $R^2 = 0.82$.

At the base resolution of $0.1\,\mathrm{cm}$ we find a mean shared variance of $R^2 = 0.56$ between the average and the individual density profiles. It can be increased by smoothing and obtains a maximum of $R^2 = 0.71$ when using a $4.3\,\mathrm{cm}$ moving average. In comparison, for 1000 randomly generated density data sets (e.g. Fig. 8), the respective stacked profiles share an average of $R^2 = 0.44$ with their components at base resolution. The maximum is $R^2 = 0.61$. We determine a $p$-value (probability of finding such high $R^2$ by chance) of $0.015$ for the measured profiles within the distribution, i.e. the high shared variance of the measured profiles is statistically significant.

## 4.2 Raw densities, isotope extrema and accumulation rates

All of the liners show at least two melt layers in the snow isotopically dating back to the summer of 2012. In addition, some liners show melt layers which are surrounded by snow dating to winter 2011/2012 or summer 2011. For an overview of all melt layers see Table 3 or Fig. 6. From the raw density profiles, we obtain Fig. 9, that shows the average densities of the top





meter and decimeter, which do not contain any prominent melt layers. The density in the top meter tends to decrease from the maximum of $332\,\mathrm{kg\,m^{-3}}$ at NEEM down to a minimum of $297\,\mathrm{kg\,m^{-3}}$ roughly $150\,\mathrm{km}$ from EGRIP before slightly increasing again. For 15 out of 18 positions the surface density is higher, nonetheless both parameters evolve similarly along the traverse.

Table 3 displays the mean annual accumulation rates along the traverse. Starting with a maximum of $225\,\mathrm{kg\,m^{-2}\,a^{-1}}$ at

NEEM the values steadily decrease down to the minimum of $115\,\mathrm{kg\,m^{-2}\,a^{-1}}$ about $100\,\mathrm{km}$ from EGRIP before slightly increasing again to $140\,\mathrm{kg\,m^{-2}\,a^{-1}}$. Comparing average values for the different years there is neither a trend nor considerable variations in the accumulation rate (cf. Table 4). However, we observe much higher differences between successive years within the same core (average change $34.67\,\mathrm{kg\,m^{-2}\,a^{-1}}$), where we mainly see alternating behaviour of high and low accumulation years.

Of the five years contained in our data, 2012 had the isotopically warmest summer for $83\%$ of the positions. At the three remaining locations (N2E_11, N2E_16 and EGRIP), the highest $\delta^{18}$O values occur in 2014. For the winters, 2014/15 was isotopically coldest in $51\%$ of the cases, 2011/12 in $19\%$ and 2010/11 in $30\%$. Regarding annual $\delta^{18}$O averages of all available positions (Table 4), we also find the highest $\delta^{18}$O values for 2012.

### 4.3 Linking accumulation, $\delta^{18}$O and density

Comparing the annual average $\delta^{18}$O values with the accumulation rates we obtain Fig. 10. Positive linear relations were fit to the data of 2012, 2013 and 2014 respectively, showing that within one year higher temperatures coincide with higher accumulation. The coefficient of determination is highest for 2012, while we have more outliers for the other two years, in particular 2013.

To relate the density with the seasonal, low frequency $\delta^{18}$O signal at NEEM, we applied a $10\,\mathrm{cm}$ running mean to the

stacked high resolution density profile in Fig. 11. On average, snow with a high $\delta^{18}$O value (considered summer snow) has a low density and the other way around. The only exception is the summer 2012, where we find high density values in summer, too.

## 5 Discussion

### 5.1 New methodology

The liner technique allows us to retrieve non-disturbed snow samples from the field and thereby conduct lab-based analysis (such as high resolution density measurements) to gain further insight in the development of physical snow properties over large distances. This is a major improvement compared to previous methods, e.g. for measuring snow density, which was so far mainly done by weighting a known volume of snow where we have a trade-off of accuracy (bulk density) and resolution (density cutters). Both, horizontal resolution and vertical depth can be adjusted to fit the needs of the respective study.

Figure 4 illustrates that we are able to align $\delta^{18}$O and density data down to small stratigraphic features very well over a distance of over $200\,\mathrm{km}$. Along the traverse, one observes a clear change in the RMSE (cf. Fig. 5) and thereby the snow



structure at the fourth liner, indicated by significantly different fitting errors. This coincides with the location where the ice divide was left eastwards and thereby the traverse entered a different accumulation regime in agreement with the drainage systems given by Zwally et al. (2011).

Furthermore the continuous depth scale agrees very well with the melt layer positions detected during the CT measurements (Fig. 6). Stratigraphic features are still well aligned over the complete traverse distance of almost 450 km. We obtain a clear picture of the layering of the snowpack along the traverse. In comparison to radar measurements, which are limited to centimeter vertical resolution but can resolve annual layers down to 12 m (Hawley et al., 2006), we can give a much more precise picture and observe small scale structures like wind crusts. In exchange we are limited to shallower depths – the maximum we plan to access in the near future are six meters in a trench at the EGRIP drilling site.

For rescaling the stacked profile to a location in the area with known annual accumulation, we obtain a linear relation of the depth factor with the ratio of accumulation rates. This is plausible, because, on average, we find linearly increasing shifts for the matching. Furthermore we do not expect significant compaction in the upper two meters of the snowpack and therefore the depth of snow from the same deposition event is determined by the accumulation rate. In addition, the relation has a high coefficient of determination for the applied linear least squares.

As the stratigraphy does not seem to change significantly along the traverse apart from the effect of the decreasing accumulation rate, we consider the profile in Fig. 7 to be representative for the whole traverse region, potentially even most of North Greenland. A statistical test using surrogate density profiles shows that the high shared variance of the measured profiles is statistically significant($p = 0.015$). Furthermore, a coeffient of determination of $R^2 = 0.56$ between the stacked and the individual profiles shows how much of the layering does reappear. Smoothing increases $R^2$ up to $0.71$ as it steadily transforms the profile to the low resolution density curve that shows seasonal behaviour (see Fig. 11) while smaller local variations vanish.

## 5.2 Temporal and regional variability of snow properties

The vast majority of melt layers is found in snow dating back to the very warm summer 2012 (Nghiem et al., 2012). Moreover, above most of the melt layers within older snow, we find clear signs of percolation (cf. Fig. 2). Therefore we assume that 2012 was the only year in the period 2010–2015 with significant melt occuring in the observed area. From Fig. 9 we can infer that on the one hand the average density of the snow in the top two meters at a certain position can already be deduced from the surface density. On the other hand the surface snow in May is among the denser ones within the year, thereby rather representing a spring or even winter signal than a sommer one (compare Fig. 11). Furthermore we are able to visually identify many layers of homogeneous density, often clearly separated by wind crusts, that thereby seem to contain snow from single deposition events.

For the accumulation rate (see Table 3) the 1964 – 2005 average of $220\,\mathrm{kg\,m^{-2}\,a^{-1}}$ determined from the NEEM ice core (Steen-Larsen et al., 2011) agrees very well with the $225\,\mathrm{kg\,m^{-2}\,a^{-1}}$ that we obtain from the corresponding snow liner. In addition, both, accumulation maps from field measurements (Bales et al., 2009) and regional climate models (Fettweis, 2007), show the same behaviour towards the East. While Table 4 shows no significant interannual changes in the average accumulation rate for the study area, we observe high fluctuations in the local annual values, a feature consistent with the strong influence of stratigraphic noise in single profiles (Muench et al., 2015). These can be explained by the accumulation of every year





compensating previous local variations in the snow surface before new structures are introduced by wind-induced drift and dunes. Nonetheless, they also might partly originate from the uncertainty of separating the years only according to the $\delta^{18}$O extrema.

In the majority of cases we find the highest isotopic summer temperatures and average $\delta^{18}$O values for 2012, underlining the exceptional warmth of this year. The values for 2014 indicate that it was still warmer than the other contained years, in particular 2010, which was formerly regarded as very warm (Harper et al., 2012). The picture for the winters is less clear. Furthermore we assume that the isotopic signal of the fresh snow from winter 2014/15 might still change.

## 5.3 Relations of density, $\delta^{18}$O and accumulation rate

We find a positive linear relationship of annual mean $\delta^{18}$O and accumulation rate (Fig. 10) with similar slopes for 2012 and 2014. This relation might partly originate from the changing surrounding conditions (e.g. elevation) along the traverse. The offset between the years could potentially be caused by the very high temperatures and the consequential surface melting in 2012 as we find the relation for 2013 to be a lot closer to 2014 than 2012. The dependence of the offset on the annual mean temperature (which is quite similar along the traverse) could explain why previous attempts to link both parameters by averaging data from several years (e.g. Weißbach et al., 2016) show less clear results.

We observe a clear anticorrelation of low resolution density and $\delta^{18}$O in Fig. 11. This agrees with the widely accepted conceptual model of Shimizu (1964) which states that snow has lower densities in summer and higher ones in winter. The high average densities in summer 2012 are caused by the prominent melt layers, superimposing the original snow density signal.

## 6 Summary and conclusions

We introduced the liner technique, that allows the very efficient retrieval of high quality samples from the upper meters of the snowpack. To support this new sampling technique, we adapted a robust fitting algorithm from accustic signal processing for the diverse data sets produced by such studies. This enables us to identify characteristic changes in the snowpack according to surrounding conditions as well as to generate a continuous depth scale using features from all available records.

To demonstrate their feasibility we applied the described methods to the upper two meters of snow along a traverse in North Greenland. We obtain a record up to May 2015 of the depths of the 2012 melt layers and sub-millimeter resolution densities. By combining these with $\delta^{18}$O measurements, that indicate temperature, we are able to reconstruct accurate accumulation rates for the years 2010 – 2014 along a distance about 400 km.

We combine isotope and density data as inputs for the matching algorithm. Thereby we are able to identify the different accumulation regimes along the traverse and resolve the continuous stratigraphy of the snow over the whole distance. This allows us to create a representative density profile for the study area, whose quality is proven by comparison with randomly generated data based on a statistical density model. The profile is available at a resolution of 0.1 cm and only has to be rescaled according to accumulation rate. Thus it is ready to act as a benchmark for model outputs or be applied for the conversion of volume to mass in remote sensing (compare e.g. Hurkmans et al., 2014).





The success of fitting density and isotope profiles over hundreds of kilometers shows that even though there is a local component in the snow stratigraphy (e.g. layer thickness, average density) the general pattern is dominated by non-local processes in North Greenland. We assume that an important factor for that is the origin of weather and precipitation as air masses dominantly move in from the West to the East (Chen et al., 1997).

We observe large interannual accumulation variations locally but almost none on average, which can be explained by the smoothing of the surface by accumulation before new surface structures are caused by dunes and drift. The exceptionally warm summer 2012 is clearly visible in the water isotope data, additionally 2014 shows the second highest summer values of $\delta^{18}$O within the study period.

   Relating the various snow properties we find a distinct anticorrelation of smoothened density and $\delta^{18}$O in accordance with
previous literature. Furthermore we deduce a positive linear relation between $\delta^{18}$O and accumulation rate, whose slope seems to be constant for the period considered while the offset varies between the years and thus might be temperature-dependent. This, however, poses the question whether models commonly used in the dating of deep ice cores (e.g. Parrenin et al., 2007, for the EPICA Dome C ice core) do correctly reconstruct accumulation rates from the $\delta^{18}$O values, especially for times with significantly differing annual mean temperatures such as glacials.

Future work should include the automatic recognition of wind crusts and layering from CT images and the application of the described methods on different scales for both Antarctica and Greenland to gain further insight into the variablity of physical properties in the snowpack.

*Author contributions.* Sepp Kipfstuhl took the samples and initiated the analysis process. Hans Christian Steen-Larsen was involved in the field planning and helped interpret the results with his expertise in the Greenland snowpack. Johannes Freitag originally established the CT
method, supervised and evaluated the isotope measurements and regularly discussed preliminary results with the main author. Olaf Eisen helped relating the results to the literature and provided insights on alternative methods. Thomas Laepple recommended underlining the results with randomly generated data and suggested possible approaches. Christoph Schaller coordinated the CT measurements, evaluated and analyzed the combined data and prepared this manuscript. It was reviewed by all coauthors.

*Acknowledgements.* The main author wants to thank the German National Merit Foundation (Studienstiftung des deutschen Volkes e.V.) for
funding his PhD project, York Schlomann for conducting the isotope measurements and the involved CIC employees for organizing and supporting the traverse.




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



**Table 1.** Measurement positions along the traverse, see also Fig. 1. The missing liner numbers (e.g. N2E_01) result from multiple samples being taken at some locations. Nonetheless, only one profile per position was used for this study.

| Position | Longitude | Latitude | Traverse kilometer |
|---|---|---|---|
| NEEM (N2E_02) | 51.06914° W | 77.444337° N | 0.00 |
| N2E_03 | 50.11° W | 77.3669° N | 24.80 |
| N2E_04 | 49.23077° W | 77.25429° N | 49.66 |
| N2E_05 | 48.170872° W | 77.120098° N | 79.76 |
| N2E_06 | 47.13806° W | 76.98195° N | 109.73 |
| N2E_07 | 46.14227° W | 76.84788° N | 138.90 |
| N2E_08 | 45.27375° W | 76.71337° N | 165.57 |
| N2E_09 | 44.78786° W | 76.52426° N | 190.03 |
| N2E_10 | 44.09225° W | 76.40034° N | 212.78 |
| N2E_11 | 43.06116° W | 76.32535° N | 241.07 |
| N2E_12 | 42.051636° W | 76.248888° N | 269.01 |
| N2E_14 | 41.16026° W | 76.1777° N | 293.92 |
| N2E_15 | 40.29929° W | 76.10455° N | 318.25 |
| N2E_16 | 39.31873° W | 76.01559° N | 346.32 |
| N2E_17 | 38.46937° W | 75.93539° N | 370.88 |
| N2E_19 | 37.69747° W | 75.85845° N | 393.48 |
| N2E_20 | 36.54374° W | 75.70614° N | 429.25 |
| EGRIP (N2E_22) | 35.985618° W | 75.629343° N | 446.83 |

**Table 2.** Fitting parameters for our adaption of the DTW algorithm.

| Property (step) | Step size ($l$) | Maximal surface offset ($s$) | Maximal bottom offset ($b$) |
|---|---|---|---|
| $\delta^{18}$O (coarse) | 3 cm | 15 cm | 75 cm |
| Density (fine) | 0.1 cm | 10 cm | 10 cm |




**Table 3.** Melt layers, the water isotopic season of origin for the surrounding snow and mean annual accumulation rates for each position. The given depths indicate the vertical center of the respective melt layer. The upper two melt layers are always located in snow from summer 2012. For the lower ones, the season of origin for the surrounding snow is given, where S indicates summer and W winter. The accumulation rates are annual mean values for all available years at the particular position.

| Position | Depth 1 [m] | Depth 2 [m] | Depth 3 [m] | Snow origin | Depth 4 [m] | Snow origin | Accumulation $[\mathrm{kg\,m^{-2}\,a^{-1}}]$ |
|---|---|---|---|---|---|---|---|
| NEEM | 1.76 | 1.84 | | | | | 224.69 |
| N2E_03 | 1.61 | 1.68 | 1.76 | S2012 | | | 193.8 |
| N2E_04 | 1.47 | 1.60 | 1.77 | W11/12 | 1.87 | W11/12 | 205.04 |
| N2E_05 | 1.35 | 1.54 | 1.67 | W11/12 | | | 171.55 |
| N2E_06 | 1.48 | 1.67 | | | | | 193.46 |
| N2E_07 | 1.37 | 1.50 | | | | | 165.38 |
| N2E_08 | 1.37 | 1.41 | | | | | 162.67 |
| N2E_09 | 1.33 | 1.42 | | | | | 155.85 |
| N2E_10 | 1.31 | 1.39 | | | | | 135.01 |
| N2E_11 | 1.21 | 1.36 | 1.50 | W11/12 | | | 137.58 |
| N2E_12 | 1.15 | 1.21 | | | | | 124.73 |
| N2E_14 | 1.12 | 1.18 | | | | | 117.30 |
| N2E_15 | 1.10 | 1.20 | | | | | 126.78 |
| N2E_16 | 1.13 | 1.16 | 1.33 | W11/12 | | | 115.06 |
| N2E_17 | 1.19 | 1.23 | 1.50 | W11/12 | | | 129.88 |
| N2E_19 | 1.13 | 1.17 | 1.42 | S2011 | | | 132.16 |
| N2E_20 | 1.35 | 1.41 | 1.48 | W11/12 | 1.61 | S2011 | 145.93 |
| EGRIP | 1.22 | 1.32 | 1.57 | W11/12 | | | 139.57 |

**Table 4.** Mean deviations of the given year from the average local annual (winter to winter) accumulation rate and $\delta^{18}$O. For each year, data from all available sites were used.

| Year | $\dot{a}$ anomaly $[\mathrm{kg\,m^{-2}\,a^{-1}}]$ | $\delta^{18}$O anomaly [‰] | Unavailable positions |
|---|---|---|---|
| 2014 | -2.66 | -0.88 | - |
| 2013 | 5.26 | -1.25 | - |
| 2012 | 3.20 | 3.64 | NEEM, N2E_06 |
| 2011 | -7.37 | -2.31 | NEEM, N2E_03-N2E_09 |



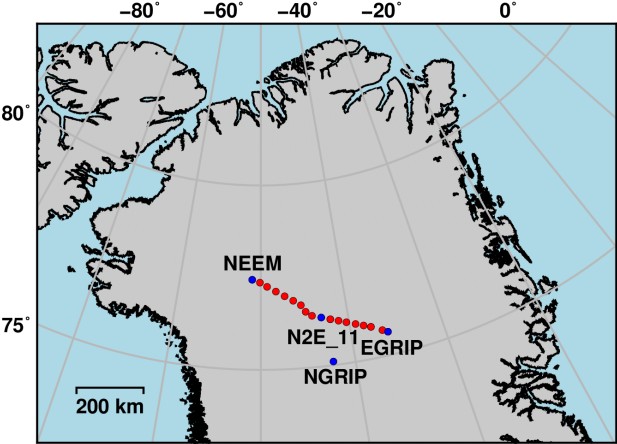

**Figure 1.** The N2E traverse route with the measurement positions according to Table 1.

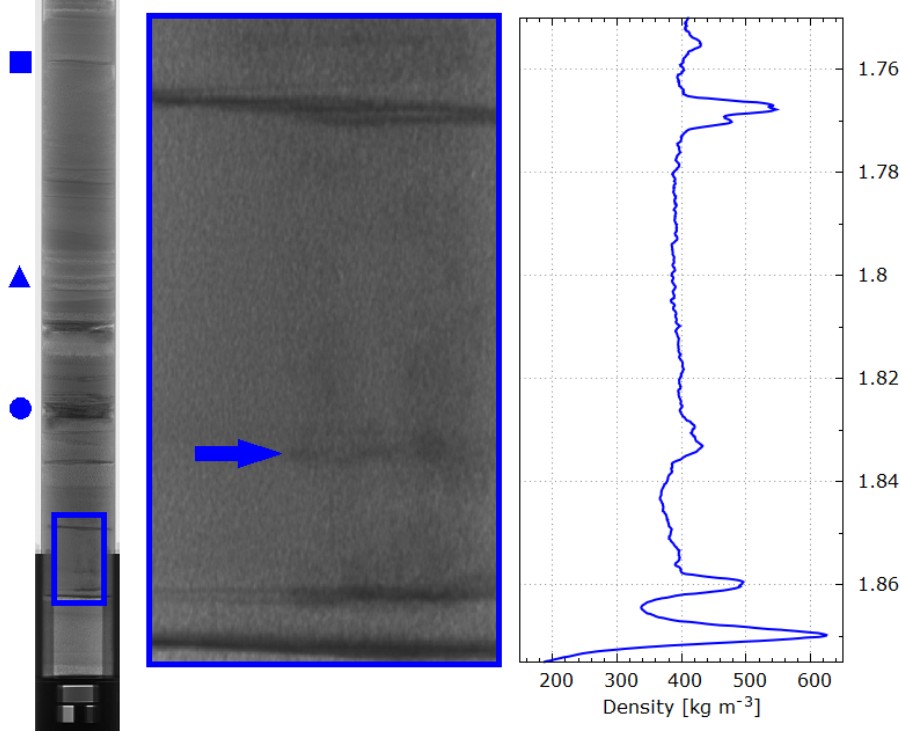

**Figure 2.** Example 2D CT image of a one meter liner (depth: $1-2\,\mathrm{m}$) and a zoomed section showing two melt layers aligned with the respective densities. In the left image a distinct density layering (e.g. blue triangle), several melt layers(e.g. blue circle) and wind crusts(e.g. blue square) are visible. Above the lower zoomed melt layer a clear percolation pattern (blue arrow) can be seen on the right hand side of the snow core.



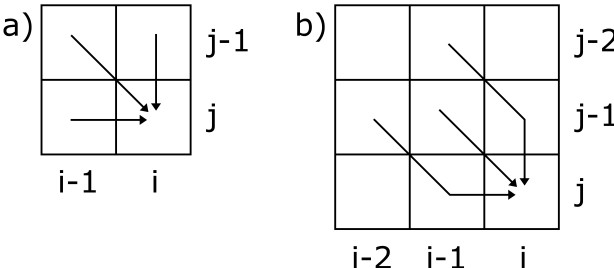

**Figure 3.** a) Basic and b) constrained stepping patterns for the DTW algorithm. The basic pattern allows for a single value to be assigned to arbitrarily many of the other data set, while for the constrained stepping each value can only be matched with one or two others.

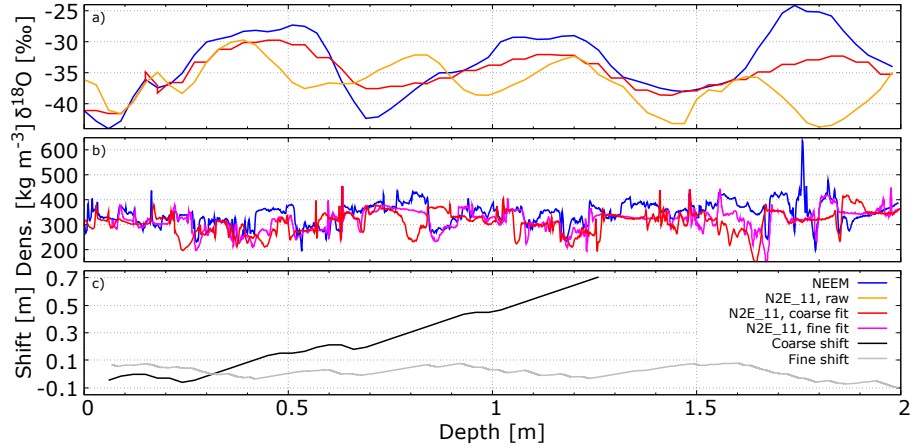

**Figure 4.** Alignment of the data from NEEM and N2E_11. a) First, the raw $\delta^{18}$O data from N2E_11 (orange) are fit to those of NEEM (blue) resulting in the red curve. b) Then, the calculated (coarse) shifts are applied to the raw N2E_11 density data to obtain the red curve as an input for a second alignment with the raw NEEM density profile (blue). We end up with the pink curve as a final result. c) The applied coarse (black) and fine (gray) shifts.




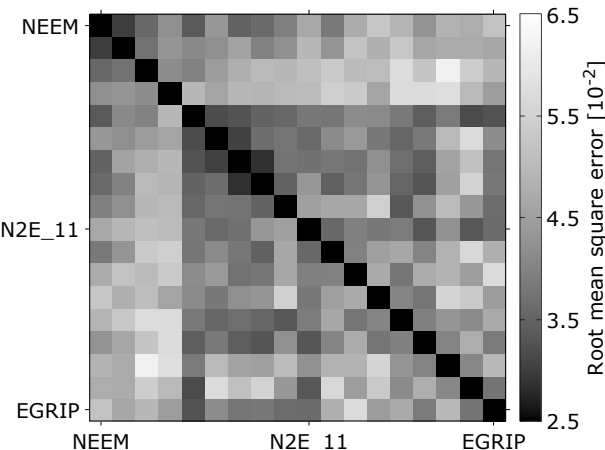

**Figure 5.** Root mean square matrix of the density alignment. The $n$'th field in the $m$'th row refers to the error of fitting data from the $n$'th and $m$'th liner. The darker the color, the lower the error and therefore the higher the agreement. Between the fourth and the fifth column (or row) a notable change in snow structure can be observed.

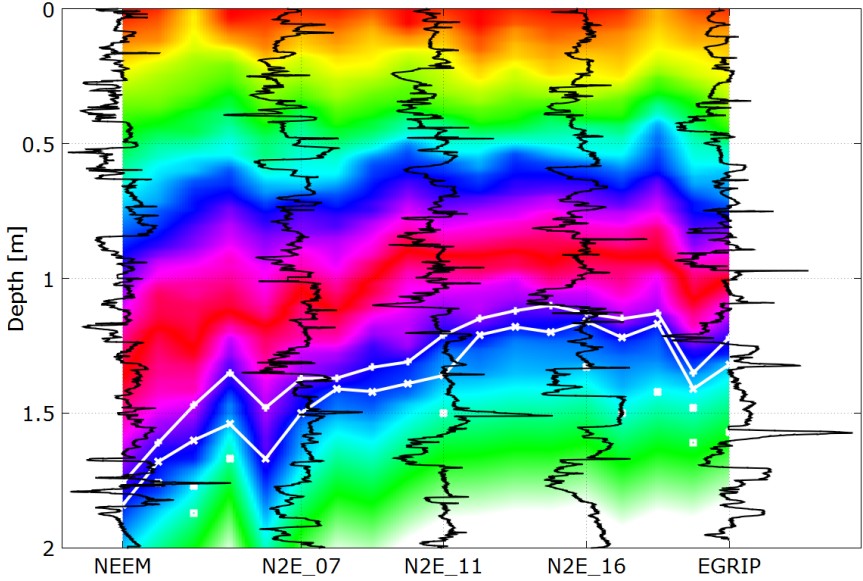

**Figure 6.** Moving depth scale, example density profiles and melt layers. A colormap was applied uniformly at the first position (NEEM) and then transformed the same way as the depths were aligned. Thus snow within the same color band was matched during the fitting process. Measured density profiles for the labeled positions are shown in black. The white lines and points indicate the melt layer positions detected from the CT scans (cf. Table 3).





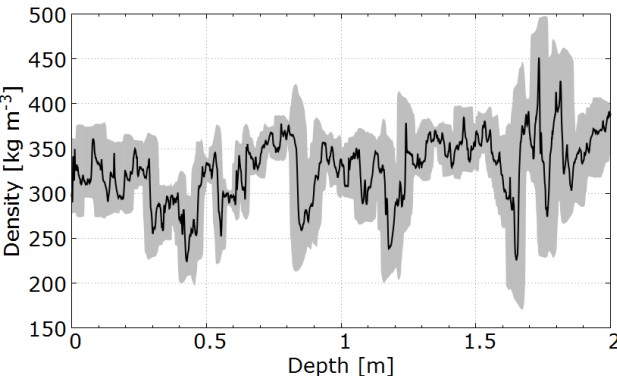

**Figure 7.** Representative density profile for the traverse region. The gray area indicates a one standard deviation error band in both $x$- and $y$-direction as there are uncertainties in the depth alignment as well as the averaged densities of all positions. Here, the depth scale was adjusted to the NEEM accumulation rate and has to be rescaled according to accumulation rate for different sites.

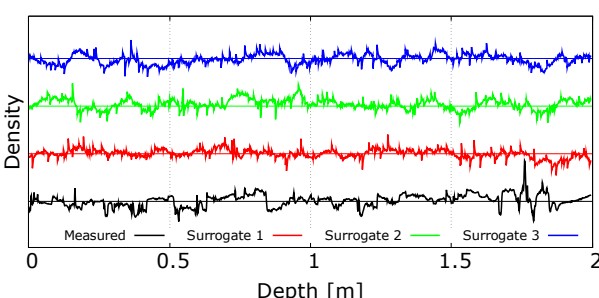

**Figure 8.** The measured density profile and three surrogates for the first position (NEEM). The random profiles are based on the seasonal $\delta^{18}O$ component of the density and have the same statistical properties as the original curve.

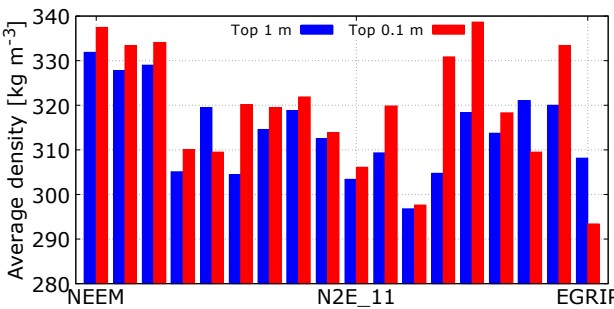

**Figure 9.** Average densities along the traverse through North Greenland (May 2015) in the top $1\,\mathrm{m}$ and $0.1\,\mathrm{m}$ derived from CT data.



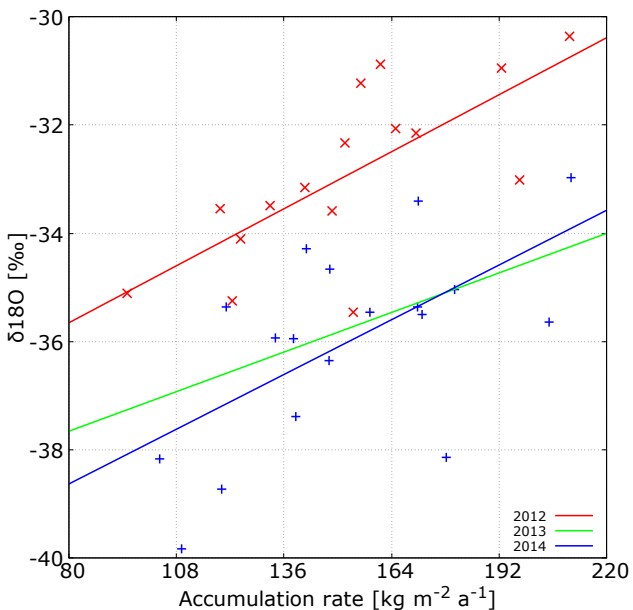

**Figure 10.** $\delta^{18}$O signal versus accumulation rate for the years 2012 – 2014. The lines were obtained by linear least squares fitting with coefficients of determination of $R^2 = 0.52$ for 2012, $R^2 = 0.27$ for 2013 and $R^2 = 0.37$ for 2014. The data points for 2013 show a few outliers and were omitted for clarity.

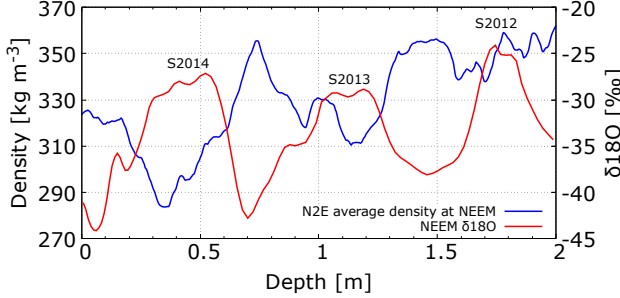

**Figure 11.** Comparison of the NEEM $\delta^{18}$O signal with the stacked density profile on the NEEM depthscale smoothed using 10 cm running means. The summer maxima for 2012 – 2014 are marked.