# Peer review of "A representative density profile for the North Greenland snowpack"

_The Cryosphere, 2016_

## Referee Comment (RC1) · Anonymous Referee #1 · 21 Jun 2016

Schaller et al. use a new technique for measuring snow that aims for pristine sampling of the top 2 meters for retrieval and analysis in the laboratory, and present a dynamic time warping (DTW) feature alignment method. The authors construct an average density profile for the North Greenland ice sheet and compare accumulation rates, melt layers and isotopic values in the area over several years. The sampling technique and CT system are very nice, the analysis is sophisticated and thoughtful. What isn't clear is how representative the product is, the density profile.

The authors include significance testing of their density alignment, which is a good idea. However, the actual numbers are marginal. The analysis uses artificial data to compare correlation coefficients for real versus fake data sets, but the resulting coefficients are not that much higher for real data. Why are the density profiles smoothed before covariance testing? Are the fake data sets also smoothed in evaluating significance? Comparing example profiles in Figure 6, and the example representative profile in Figure 7 with its substantial one-sigma confidence interval, it's hard for the reader to judge what is being captured or how useful it is. The bottom two-thirds of the representative profile in Figure 7 is consistent with a straight line.

This is not the first time that DTW or other speech/biometric processing approaches have been adapted for stratigraphic alignment of environmental records, or even ice core records, and Schaller et al. may benefit by referring to these and other probabilistic approaches. DTW is most effective when aligning time series containing prominent features that are highly similar. Since the goal (based on the manuscript title) is an average profile, it would be useful to check consistency between the various record alignment combinations, e.g., the features matched between N2E_04 and N2E_05 with NEEM, should also match between N2E_04 and N2E_05 with each other.

Technical corrections, typos and style:

Abstract, line 5: suggest striking "based"

Section 2.2, line 29: "...a worldwide unique..." suggest striking "worldwide"

Section 3.1, line 22: "...could be fit to arbitrary many..."

Section 4.1, line 11.

---

## Referee Comment (RC2) · Anonymous Referee #2 · 12 Jul 2016

General comments:

This paper presents a new technique for efficiently retrieving shallow snow and firn cores from polar regions. Those cores can then be returned to a lab for high-resolution analysis using a micro-CT scanner. These types of measurements are needed to better understand the evolution of firn, which in turn will lead to more accurate estimations of mass-balance changes on the ice sheets. The authors apply a technique that was developed for speech recognition, Dynamic Time Warping (DTW), to analyze changes in snow and firn properties along a 450-km traverse in northern Greenland. Additionally, the authors examine variability in annual accumulation rates and relationships between water isotopes (a temperature proxy) and accumulation rates.

The paper makes a valuable contribution to the glaciological community and will be

of particular interest to those who study snow and firn related surface-mass-balance processes and ice-core delta age estimation. The "liner" technique combined with DTW could easily be adopted by the snow-hydrology and snow-avalanche communities to investigate snow properties on smaller spatial scales. I have 2 general comments and numerous specific comments that I would like to have addressed before publication.

"Matching" snow and firn properties. The authors use DTW to "match" the firn properties along the traverse. The first step of the alignment is to match the d18O data, which identifies snow/firn from a particular summer or winter. This seems to me to be the most valuable use of the DTW technique because it gives an idea of how accumulation is varying seasonally and annually over a large distance (temporal and seasonal variabililty).

Their next step is to align high-resolution density features in the snow/firn. However, I am left unsure what information this high-resolution matching or alignment is providing. What is the end goal in aligning the high-resolution density data? Is it to track layers deposited during individual weather events? Or to provide a common depth-age profile along the traverse? Related to this question: What does the "fine fit" in Figure 4b mean physically, and why is that a useful metric? Likewise, what is the physical meaning of the color bands in Figure 6? Would those be layers of snow with the same age?

I believe that using DTW on the high-resolution density data includes an a priori implicit assumption that stratigraphic features (layers) and are continuous (or at least correlated) over hundreds of kilometers, but the authors have not convinced me that this is or should be true. Why do you expect the depth-density profiles to be related? Does this argument hold up if this assumption is not true? Recent work by Proksch and others (2015, e.g. Fig. 12) showed significant stratigraphic variability in the near-surface snow in Antarctica. I would expect some amount of coherence on the 10's-of-kilometers scale, but it is surprising to hear that stratigraphic features (and coherence in density) persist over hundreds of kilometers and over a divide, where temperatures and accumulation rates vary on daily to annual time scales. If the authors are assuming that layers persist over these distances, at what layer resolution would they expect this assumption to break down? Can you be confident that the algorithm is matching real layer correlations and not just recognizing stochastic layering that all happens to fall near some mean density?

The authors do discuss verification of their method using surrogate density profiles. However, I do not follow their reasoning – this could be a place to clarify their language.

Ultimately, the authors do not make a strong case to me that the layers they are fitting are spatially extensive and not stochastic noise. I request that the authors justify the assumption that the layers are spatially extensive. Additionally, they should clarify the language of what the alignment using high-resolution depth-density data means. An example of somewhere to clarify: Page 5 Line 4 says, "... the continuous depth scale agrees..." Perhaps specifying what a continuous depth scale means would help me understand – is that a continuous depth-age scale? Alternatively, the authors could focus on the DTW using the d18O data.

Uncertainty and application to mass balance. The authors point out in the introduction the importance of knowing firn properties for mass balance calculations, and they derive a representative depth-density profile. How much uncertainty is associated with using this representative profile? I suggest that it would be useful to compare the representative depth-density profile to measurements and model predictions. A metric of interest for the mass-balance community is the depth-integrated porosity (DIP), or the amount of air in the snow and firn. I think it would be a useful exercise to compare the DIP that is observed in the cores to the DIP that is predicted by the representative profile. Additionally, it could be compared to the DIP predicted by assuming some constant density for the top 2 m and perhaps to density profile predicted by a firn-densification model.

Specific comments:

- Page 2, Line 4: thereby measurements of what? - P2, L10: what individual parameters? - P2, L20: How do you know in which cases the snow might be compacted? - P2, L28: Are you confident that no metamorphism occurs during transport, e.g. due to temperature gradients? - P2, L31: "Amongst others,..." amongst other what? Other corrections? If so, state what those are. - P3, L24: a shift in what? - P3, L26: what do you mean by event? - P3, L26: what do you mean by align? (related to general comment above) Snow of a certain age? - P3, L30 – P4: DTW is a complicated concept to read through for the first time – perhaps you can provide an example in this section – e.g. what "assigning the values of" means, what "proceeding through the matrix" means, etc. - P5, L3: Combine all of the available information: are you using anything besides d18O and density? - P5, L9/Table 2: How did you come up with you maximal/minimal offset values? (Should that be maximum/minimum?) - P5, L12: These aren't really continuous, are they? You have discrete measurements from every ~25 km along the traverse. How do you interpolate between those? - Section 3.2: Are the statistics informing the creation of the surrogate density profiles taken from the bulk of all geographic locations (i.e. sigma_base is the standard deviation from all sites) or from single sites? - P5, L24: Do you calculate the surrogate profiles for each site individually using that site's statistical properties, or the bulk statistical properties of all sites? - P6, L12: How do you define a "layer"? - P7, L9: How is 0.1m chosen as the maximum allowed shift? - P7, L10: What does "all combinations of 2 liners" mean? Do you mean that you are comparing the 2 meters of data from each site to each other site? - P7, L11-12 and P9, L1: Can you further elaborate on why that change occurs? Going over the divide, you lose some coherence, but not all? Which signals are lost going over the divide, and which are maintained? Is there also a change at the $\frac{1}{2}$ way mark? - P7, L15: It is unclear to me: was the representative density profile created by stacking the raw depth-density data, or by using the "aligned" profile from DTW? - P8, L9: Do the high and low accumulation years correlate spatially? I.e. does a high-accumulation year at NEEM also mean high accumulation at EGRIP? - P8, L12: Is this spread in isotopic coldest year surprising? Are those data corroborated by reanalysis (e.g. RACMO) data? - P8, L18 and Figure 10: What is the source of those

outliers? Sampling/instrument error? Please elaborate on how you identified them as such. - P8, L20-21 (and P10,L16): I think it would be appropriate to elaborate on why the summer snow has a lower density. - P9, L12: Why do you not expect significant compaction? Reference or justify. - P10, L6: Why is the winter picture less clear? - P10, L9: You show that warmer sites have more accumulation. For a given site, does a warmer winter, summer, or year correlate with higher accumulation at that site? - Figure 3: Does not clarify the constraints on stepping to me. - Figure 6: The densities that are plotted do not have labels, scales, or units. Are those the centerline values and scales same for each? It might be helpful to mark the mean density and standard deviation of each of depth-density profile. - Figure 7: What are your x and y directions? Can you elaborate on how you get the standard deviation error band, e.g. is it comparing the raw data from each site to the representative profile?

Technical corrections: - Numerous places in text the authors use vague language: e.g. "profile" (depth-density profile, depth-age, density-age profile, "depth profile" is still vague), "position" (position could refer to some point on the firn core rather than a geographic location, and I suggest a change to "site" or similar) - Throughout: The authors use the language "the liners show" or similar (e.g. P5L12); the "liners" are the instruments/tools used to gather their data and are not what actually show anything. I suggest language such as "the data from each of the sites show" - There are numerous instances throughout the manuscript where (1) commas are misused or lacking and (2) hyphens are needed. - Several places in the text change tense (past vs. present, e.g. section 2.1) and voice (active vs. passive). I suggest choosing one. - Page 1, Line 5: empirical based empirically-based - Page 1, Line 11: impact impacts - Page 2, L1: causing creating - P2, L25-26: probing sampling, "that technique" "the liner technique" - P3, L3: measurement time increases with resolution? (rather than accuracy) - P3, L5-6: "Then, the raw...CT images." Unclear sentence - P3, L7: weight mass - P3, L15-16: clarify that is it 3-5 years worth of accumulation contained in the 2-m sample; specify "winter-to-winter accumulation rates" - P3, L21: renowned well-known - P4, L23: what does the "maximal ratio of the respective accumulation rates" mean? Repective to
what? Two sites next to one another? - P6, L3: Perhaps use $z_i$ since you are talking about depth. The x dimension (to me) indicates a direction on the surface (e.g. along your traverse). - P7, L15: Change to "The previously-calculated depth-scale density records were stacked to obtain..." - P8, L6: Increasing to 140 where? - P9, L1: fourth liner? Do you mean location/site? - P9, L15: The statistics in this paragraph were already reported on page 7; did you intentionally do that? - P9, L21: is are, summer of 2012 - P9, L27: sommer summer - P10, L20: accustic acoustic - Figure 8: y-axis does not have scale or units labeled - there are numerous instances of typos and challenging-to-read sentence structure that I have not indicated here; I recommend having a copy editor review the manuscript for those.

Reference:

Proksch, M., Löwe, H., & Schneebeli, M. (2015). Density, specific surface area, and correlation length of snow measured by high‐resolution penetrometry. Journal of Geophysical Research: Earth Surface, 120(2), 346-362.
* * *

---

## Author Comment (AC1) · 7 Aug 2016

We would like to thank the two anonymous referees for their critical and helpful comments. We have responded to all of them below and tried to adress as many as possible to significantly improve the manuscript. For the few cases where we did not follow the reviewers' suggestions, we discuss the reasons for our decision. The referee comments are displayed in italics, followed by our responses in normal font.

[Figure]

**Anonymous referee #1**

*Schaller et al. use a new technique for measuring snow that aims for pristine sampling of the top 2 meters for retrieval and analysis in the laboratory, and present a dynamic time warping (DTW) feature alignment method. The authors construct an average density profile for the North Greenland ice sheet and compare accumulation rates, melt layers and isotopic values in the area over several years. The sampling technique and CT system are very nice, the analysis is sophisticated and thoughtful. What isn't clear is how representative the product is, the density profile.*

*The authors include significance testing of their density alignment, which is a good idea. However, the actual numbers are marginal. The analysis uses artificial data to compare correlation coefficients for real versus fake data sets, but the resulting coefficients are not that much higher for real data.*

Indeed, the numbers are not that much higher. However the main cause for this fact is that we compare with the most realistic surrogates that we were able to create. The artificial datasets consist of the real $\delta^{18}O$ profiles and a density signal based on the seasonal cycle, a $\delta^{18}O$-density relation and a statistical model with three components for the stratigraphy. If we use a simple statistical method (e.g. autoregressive processes) we can generate much more impressive numbers, but instead we wanted to show that even in comparison to surrogates with the maximum amount of real information there is a small, but significant difference. We tried to clarify this in the manuscript by adding more details in Chapters 3.2 and 5.1.

*Why are the density profiles smoothed before covariance testing? Are the fake data sets also smoothed in evaluating significance?*

As can be seen in Chapter 4.1, last paragraph, we mainly work at the base resolution of [0.1]cm for covariance testing, which corresponds to no smoothing. We only wanted to provide the information that the shared variance increases with smoothing, which can be explained by the steady transformation into a seasonal signal. Apart from that statement, all of the numbers in

this paragraph refer to the base resolution and are indicated to do so.

*Comparing example profiles in Figure 6, and the example representative profile in Figure 7 with its substantial one-sigma confidence interval, it's hard for the reader to judge what is being captured or how useful it is. The bottom two-thirds of the representative profile in Figure 7 is consistent with a straight line.*

In order to improve the understanding of the presented content, the following has been added to the discussion of Figure 7 in Chapter 5.1: "For the given error band, there is an overlap of uncertainty in the depth alignment ($x$-direction) with the uncertainty in density ($y$-direction). The former is mainly caused by the variability of the snow mass accumulated from a single deposition event. Regarding the latter, the average density of the snowpack greatly varies as can be seen in Fig. 9. Thus, for the second meter, even though it is contained in the uncertainty band, we do not expect a straight line, but rather an alternation of high and low density layers similar to the upper meter."

*This is not the first time that DTW or other speech/biometric processing approaches have been adapted for stratigraphic alignment of environmental records, or even ice core records, and Schaller et al. may benefit by referring to these and other probabilistic approaches.*

Three additional references have been added to Chapter 3.1 to provide an adequate overview of previous research - a detailed review of the DTW method (Senin, 2008), an example of its application in polar science (remote sensing of ice floes, McConnell et al., 1991) and a recent example for a different approach to align physical properties within the snowpack (Hagenmuller and Pilloix, 2016).

*DTW is most effective when aligning time series containing prominent features that are highly similar. Since the goal (based on the manuscript title) is an average profile, it would be useful to check consistency between the various record alignment combinations, e.g., the features matched between N2E_04 and N2E_05 with NEEM, should also match between N2E_04 and N2E_05 with each other.*

We apologize, this has actually been checked but not stated. "There were no notable differences when another location (e.g. EGRIP) was chosen as the reference or the fitting was done consecutively." was added to Chapter 4.1.

*Technical corrections, typos and style:*

*Abstract, line 5: suggest striking "based"*

Done.

*Section 2.2, line 29: "...a worldwide unique..." suggest striking "worldwide"*

Done.

*Section 3.1, line 22: "...could be fit to arbitrary many..."*

Changed to "a single value of one data set could be fit to arbitrarily many of the other."

*Section 4.1, line 11.*

Removed extra "of", replaced "fit" by "fitted" and "liners" by "profiles".

**Anonymous referee #2**

*General comments: This paper presents a new technique for efficiently retrieving shallow snow and firn cores from polar regions. Those cores can then be returned to a lab for high-resolution analysis using a micro-CT scanner. These types of measurements are needed to better understand the evolution of firn, which in turn will lead to more accurate estimations of mass-balance changes on the ice sheets. The authors apply a technique that was developed for speech recognition, Dynamic Time Warping (DTW), to analyze changes in snow and firn properties along a 450-km traverse in northern Greenland. Additionally, the authors examine variability in annual accumulation rates and relationships between water isotopes (a temperature proxy) and accumulation rates. The paper makes a valuable contribution to the glaciological community*

*and will be of particular interest to those who study snow and firn related surface-mass-balance processes and ice-core delta age estimation. The "liner" technique combined with DTW could easily be adopted by the snow-hydrology and snow-avalanche communities to investigate snow properties on smaller spatial scales. I have 2 general comments and numerous specific comments that I would like to have addressed before publication.*

*"Matching" snow and firn properties. The authors use DTW to "match" the firn properties along the traverse. The first step of the alignment is to match the d18O data, which identifies snow/firn from a particular summer or winter. This seems to me to be the most valuable use of the DTW technique because it gives an idea of how accumulation is varying seasonally and annually over a large distance (temporal and seasonal variabililty). Their next step is to align high-resolution density features in the snow/firn. However, I am left unsure what information this high-resolution matching or alignment is providing. What is the end goal in aligning the high-resolution density data? Is it to track layers deposited during individual weather events? Or to provide a common depth-age profile along the traverse?*

The goal is to track features (melt layers, wind crusts as well as significant changes in density at the borders of snow layers from different "events") to learn about their spatial extent and variability. Then, the fact that we are able to do so, enables us to provide a common depth-age profile and construct a representative density profile for the traverse region (cf. Chapter 3.1). The profile can be rescaled to any location of known accumulation. Amongst others, it may be applied as a benchmark for snowpack models or for the detection of strong density gradients as potential reflectors in remote sensing (for further details, see Chapter 6).

*Related to this question: What does the "fine fit" in Figure 4b mean physically, and why is that a useful metric?*

The mass accumulated by a certain event is strongly influenced by wind speed and direction (Fisher et al., 1985, cited), which are partially coherent over the region of interest (Chen et al., 1997, cited). Due to diffusion, this information might be lost post-depositionally in the $\delta^{18}O$ signal, which also has a pretty low resolution compared to the vertical extent of features (e.g.
melt layers) in the snow (cf. Chapter 3.1). Thus fitting the seasonal $\delta^{18}O$ signal provides a coarse "age"-alignment of the snow, but no "feature"-alignment. Densities, on the other hand, are available at a much higher resolution and less likely to be influenced post-depositionally. This make the second ("fine") fitting step (see also, Results, 4.1, reference to Figure 4) necessary and useful.

*Likewise, what is the physical meaning of the color bands in Figure 6? Would those be layers of snow with the same age?*

The color bands do not represent physical layers, but snow of potentially the same origin and thereby approximately the same age. This is described in the figure caption: "... A colormap was applied uniformly at the first position (NEEM) and then transformed the same way as the depths were aligned. Thus snow within the same color band was matched during the fitting process..."

*I believe that using DTW on the high-resolution density data includes an a priori implicit assumption that stratigraphic features (layers) and are continuous (or at least correlated) over hundreds of kilometers, but the authors have not convinced me that this is or should be true. Why do you expect the depth-density profiles to be related? Does this argument hold up if this assumption is not true? Recent work by Proksch and others (2015, e.g. Fig. 12) showed significant stratigraphic variability in the near-surface snow in Antarctica. I would expect some amount of coherence on the 10's-of-kilometers scale, but it is surprising to hear that stratigraphic features (and coherence in density) persist over hundreds of kilometers and over a divide, where temperatures and accumulation rates vary on daily to annual time scales. If the authors are assuming that layers persist over these distances, at what layer resolution would they expect this assumption to break down? Can you be confident that the algorithm is matching real layer correlations and not just recognizing stochastic layering that all happens to fall near some mean density? The authors do discuss verification of their method using surrogate density profiles. However, I do not follow their reasoning – this could be a place to clarify their language.*

Our plan was to identify features in consecutive profiles and steadily expand a continuous depth alignment (as in Figure 6). Thus, the a priori assumption of our work was being able to track stratigraphic features over [20-30]km in agreement with your expectations. It was not our initial intention to trace layers over hundreds of kilometers. We started by fitting the profiles consecutively, which does not significantly change the results. A sentence with this statement has been added to Chapter 4.1. Potential reasoning for coherence in the density profiles over larger distances is given (e.g. predominant origin of weather and precipitation - see Chapter 6, Line 13-16). Regarding the ice divide, it is visible in the RMSE as discussed (cf. Figure 5; Chapter 5.1, L 4-7). As we mentioned above, we do not aim to identify "layers" in a physical sense, but significant changes in snow properties. Therefore there is no "resolution" where the fitting would break down, apart from when features become smaller than the base resolution of our density alignment (i.e. [0.1]cm). We further used surrogate density profiles to test whether the increased shared variance after the fine tuning step could be explained by chance. The results show that the amount of shared variance for the measured profiles is statistically significant, which underlines that we are not just recognizing stochastic layering.

The mentioned paper shows data from Kohnen Station, which is located in an East Antarctic low-accumulation region (64 mmWE/a) and thus comparison with our traverse (115-225 mmWE/a) is not straightforward. Indeed, there is significant wind scouring (possibly causing hiatuses of the complete mass accumulated by a single event) at Kohnen (compare e.g. Muench, 2015, cited).

*Ultimately, the authors do not make a strong case to me that the layers they are fitting are spatially extensive and not stochastic noise. I request that the authors justify the assumption that the layers are spatially extensive. Additionally, they should clarify the language of what the alignment using high-resolution depth-density data means. An example of somewhere to clarify: Page 5 Line 4 says, " ... the continuous depth scale agrees ... " Perhaps specifying what a continuous depth scale means would help me understand – is that a continuous depth-age scale? Alternatively, the authors could focus on the DTW using the d18O data.*

The statistical verification shows that there is a significantly increased shared variability between the real density profiles compared to surrogate data (which we tried to design as realistic as possible, e.g. the original $\delta^{18}O$ profiles were used). Thus we provided evidence for spatial coherency of the density over hundreds of kilometers. Further details have been added to Chapters 3.2 and 5.1 to clarify this, see also first answer to RC #1. We tried to put more emphasis on showing the value of having the densities for high-resolution alignment and apologize for the misleading term "continous/moving depth scale", that was replaced by "depth alignment" (e.g. the color bands in Figure 6). Apart from statistical analyses (as already carried out), proving the existence of spatially extensive layers would require an extremely dense sampling of the stratigraphy, which is not achievable. Indirect methods, such as high-resolution shallow radar (e.g. Hawley et al., 2006, cited), indicate the persistence of layers over 10's to 100's of kilometers, but do not provide the vertical resolution to validate layer structure on the same scale as our approach.

*Uncertainty and application to mass balance. The authors point out in the introduction the importance of knowing firn properties for mass balance calculations, and they derive a representative depth-density profile. How much uncertainty is associated with using this representative profile? I suggest that it would be useful to compare the representative depth-density profile to measurements and model predictions. A metric of interest for the mass-balance community is the depth-integrated porosity (DIP), or the amount of air in the snow and firn. I think it would be a useful exercise to compare the DIP that is observed in the cores to the DIP that is predicted by the representative profile. Additionally, it could be compared to the DIP predicted by assuming some constant density for the top 2 m and perhaps to density profile predicted by a firn-densification model.*

An error band for the representative profile is given in Figure 7, further explanation regarding uncertainties has been added to Chapter 5.1. We have discussed this point and unfortunately do not see a considerable merit in calculating the DIP for two meters of snow, where we only expect marginal differences. To our knowledge it is a parameter that is rather interesting with respect to the whole firn column. In addition, the DIP for a density profile will only be determined by its average value – the formula is $DIP = 2\text{m} \cdot (1 - \frac{\rho_{avg}}{\rho_{ice}})$. Instead, we see

the main advantage of a high-resolution density profile for remote sensing in the opportunity to determine significant density contrasts that can cause strong reflections (such as the 2012 melt layers). To clarify this in the manuscript, we also changed the respective sentence in Chapter 6 to "Thus it [the representative profile] is ready to act as a benchmark for snowpack models or be applied for the conversion of volume to mass and the detection of strong density gradients as potential reflectors in remote sensing." Furthermore, there is no significant firn-densification in the upper two meters (compare sample density curves, e.g. Figure 6).

*Specific comments: - Page 2, Line 4: thereby measurements of what?*

Corrected.

*- P2, L10: what individual parameters?*

Density, $\delta^{18}O$ and accumulation rate. Added.

*- P2, L20: How do you know in which cases the snow might be compacted?*

The tube has no lid or similar. Thus, as you push it into the snow, at some point the top of the liner will be parallel to the surrounding snow surface. If the snow inside the liner is not, it has been compacted.

*- P2, L28: Are you confident that no metamorphism occurs during transport, e.g. due to tem-perature gradients?*

We transport the samples with the minimal number of transitions at a constant (low) tempera-ture, the same way it is done for ice cores (e.g NEEM). This minimizes the effects of isotopic diffusion and potential metamorphism (e.g. there is no long exposition to temperature gradi-ents). Currently we do not know about any significant impact of such transport on the conducted measurements (2D density profiles, $\delta^{18}O$). For example, a comparison of discrete density mea-surements on a trench wall close to Kohnen station and CT density measurements of liners transported to Bremerhaven showed good agreement.

*- P2, L31: "Amongst others, ..." amongst other what? Other corrections? If so, state what*

*those are.*

A more detailed description of the AWI-Ice-CT and its measurement procedure has been provided in previous publications (e.g. Freitag et al., 2013, cited). This has been clarified in the manuscript.

*- P3, L24: a shift in what?*

"in depth" added.

*- P3, L26: what do you mean by event?*

"deposition" added.

*- P3, L26: what do you mean by align? (related to general comment above) Snow of a certain age?*

"of the same origin" added.

*- P3, L30 – P4: DTW is a complicated concept to read through for the first time – perhaps you can provide an example in this section – e.g. what "assigning the values of" means, what "proceeding through the matrix" means, etc.*

We added an introductory reference to the manuscript as DTW is already a well-described method.

*- P5, L3: Combine all of the available information: are you using anything besides d18O and density?*

No. Removed "all".

*- P5, L9/Table 2: How did you come up with you maximal/minimal offset values? (Should that be maximum/minimum?)*

Maximum, yes. We added the sentence "The maximum allowed offsets for the coarse fitting have been chosen according to the measured height of variations in the snow surface (e.g. dunes) and the maximum ratio of estimated accumulation rates. In the second step we allow for fine

tuning up to the maximum remaining shift, that was manually identified by aligning the 2012 melt layers." to Chapter 3.1.

*- P5, L12: These aren't really continuous, are they? You have discrete measurements from every 25 km along the traverse. How do you interpolate between those?*

The term "continous" refers to being able to connect the profiles here. We do not use interpolation as part of our analysis, but if we interpolate for visual purposes (e.g. Figure 6) it is done linearly. This has been indicated in the manuscript.

*- Section 3.2: Are the statistics informing the creation of the surrogate density profiles taken from the bulk of all geographic locations (i.e. sigma_base is the standard deviation from all sites) or from single sites?*

From single sites. "independently" added to the first paragraph of 3.2 to clarify.

*- P5, L24: Do you calculate the surrogate profiles for each site individually using that site's statistical properties, or the bulk statistical properties of all sites?*

Added another "For each site" in the third paragraph.

*- P6, L12: How do you define a "layer"?*

Replaced by "snow of similar properties".

*- P7, L9: How is 0.1m chosen as the maximum allowed shift?*

See previous question/answer regarding maximum allowed shifts (P5, L9/Table 2).

*- P7, L10: What does "all combinations of 2 liners" mean? Do you mean that you are comparing the 2 meters of data from each site to each other site?*

Yes. Replaced by "all combinations of profiles from two sites".

*- P7, L11-12 and P9, L1: Can you further elaborate on why that change occurs? Going over the divide, you lose some coherence, but not all? Which signals are lost going over the divide, and which are maintained? Is there also a change at the 1/2 way mark?*
Yes, we lose some coherence in the divide area, but not all. To us, it is not straightforward to talk about lost or maintained "signals" here as there is no seperate "signals" identified in the matching process but whole profiles compared. The idea of Figure 5 was to provide an overview, which visualizes a change in snow structure or stratigraphy at a position that coincides with the area where the divide was left. There could be a change at the 1/2 way mark in Figure 5, but the differences between N2E_11 and the neighboring sites might be misleading. However, it is less prominent than the one at the ice divide. In order to be more precise, the caption now reads "The most notable change in snow structure can be observed between the fourth and the fifth column (or row)."

*- P7, L15: It is unclear to me: was the representative density profile created by stacking the raw depth-density data, or by using the "aligned" profile from DTW?*

The respective sentence now reads "Using the previously calculated depth alignment, density records were stacked to obtain a representative density profile", which should clarify the aligned profiles were stacked.

*- P8, L9: Do the high and low accumulation years correlate spatially? I.e. does a high-accumulation year at NEEM also mean high accumulation at EGRIP?*

Previous sentence: "Comparing average values for the different years there is neither a trend nor considerable variations in the accumulation rate (cf. Table 4)." There is no spatial correlation of low and high accumulations, as this would cause considerable variations of the annual averages.

*- P8, L12: Is this spread in isotopic coldest year surprising? Are those data corroborated by reanalysis (e.g. RACMO) data?*

We feel that including reanalysis data to discuss this point would be beyond the scope of our manuscript. This information is solely provided for the interested reader.

*- P8, L18 and Figure 10: What is the source of those outliers? Sampling/instrument error? Please elaborate on how you identified them as such.*

We apologize, the term "outlier" was incorrectly used here and has been replaced/avoided. The

large spread in 2013 cannot be explained by sampling or instrumental errors.

*- P8, L20-21 (and P10,L16): I think it would be appropriate to elaborate on why the summer snow has a lower density.*

Added "The main causes given are the increased packing due to stronger winds in winter and the larger size of precipitation particles in summer." in 5.3.

*- P9, L12: Why do you not expect significant compaction? Reference or justify.*

"Densification" would have been the more precise term here. We would be able to observe it in the density profiles (increasing average value). Sentence changed to "Furthermore we do not observe significant densification ..."

*- P10, L6: Why is the winter picture less clear?*

Chapter 4.2, last paragraph. Different years have been coldest for different sites. "Furthermore" replaced by "Indeed" to better link to the following sentence (surface signal might still change).

*- P10, L9: You show that warmer sites have more accumulation. For a given site, does a warmer winter, summer, or year correlate with higher accumulation at that site?*

We do not observe such correlation, the amount of snow accumulated in a certain year seems to mainly be determined by the surface variations.

*- Figure 3: Does not clarify the constraints on stepping to me.*

"Here usage of cell $[i, j]$ refers to $\mathbf{S}[i]$ being assigned to $\mathbf{T}[j]$." added to 3.1, after reference to Figure 3.

*- Figure 6: The densities that are plotted do not have labels, scales, or units. Are those the centerline values and scales same for each? It might be helpful to mark the mean density and standard deviation of each of depth-density profile.*

Caption of Figure 6 changed to "In black, measured density profiles for the labeled positions are shown at the same scale, centered around their respective mean values." We have tried adding

[Figure]

more information regarding the example density profiles to the plot but fear that it will decrease the clarity and thus make it more difficult for the reader to grasp the essential information. In addition, there are other plots (e.g. Figure 4) showing densities with scale and unit.

*- Figure 7: What are your x and y directions? Can you elaborate on how you get the standard deviation error band, e.g. is it comparing the raw data from each site to the representative profile?*

The following has been added to the discussion of Figure 7 in Chapter 5.1: "For the given error band, there is an overlap of uncertainty in the depth alignment ($x$-direction) with the uncertainty in density ($y$-direction). The former is mainly caused by the variability of the snow mass accumulated from a single deposition event. Regarding the latter, the average density of the snowpack greatly varies as can be seen in Fig. 9. Thus, for the second meter, even though it is contained in the uncertainty band, we do not expect a straight line, but rather an alternation of high and low density layers similar to the upper meter."

*Technical corrections: - Numerous places in text the authors use vague language: e.g. "profile" (depth-density profile, depth-age, density-age profile, "depth profile" is still vague), "position" (position could refer to some point on the firn core rather than a geographic location, and I suggest a change to "site" or similar)*

"Position" replaced by "site" or "location". Regarding "profile": The term "depth profile" is not used in the manuscript. We did use the terms "density profile", "$\delta^{18}O$ profile" and "isotope profile", which implies depth is the second parameter when talking about snow or ice cores. The term "profile" is solely used when the respective property is clear from the context or no specific property is adressed. We hope that this satisfies the reviewer's comment.

*- Throughout: The authors use the language "the liners show" or similar (e.g. P5L12); the "liners" are the instruments/tools used to gather their data and are not what actually show anything. I suggest language such as "the data from each of the sites show"*

The mentioned sentence now reads "For all sites we find at least two melt layers in the snow

isotopically dating back to the summer of 2012." Similar language has been changed throughout the manuscript.

*- There are numerous instances throughout the manuscript where (1) commas are misused or lacking and (2) hyphens are needed.*

We reread the manuscript carefully and tried to follow the TC guidelines as closely as possible. Please note that these partly differ from the guidelines of other publishers or standard dictionaries.

*- Several places in the text change tense (past vs. present, e.g. section 2.1) and voice (active vs. passive). I suggest choosing one.*

2.1. was written in present tense on purpose, in order to describe a new technique. It has been updated to past for consistency now.

*- Page 1, Line 5: empirical based empirically-based*

"based" was removed.

*- Page 1, Line 11: impact impacts*

Both seems possible. We do not see why "impacts" would be better here.

*- Page 2, L1: causing creating*

Both possible, wanting to be precise the warm days did not really "create" the melt layers though.

*- P2, L25-26: probing sampling, "that technique" "the liner technique"*

"Probing" replaced by "sampling", "that technique" removed.

*- P3, L3: measurement time increases with resolution? (rather than accuracy)*

Changed, we tried to avoid too many repetitions of "resolution" and replaced the next instance by "pixel size" instead.
- *P3, L5-6: "Then, the raw ... CT images." Unclear sentence*

Extra "the" removed.

- *P3, L7: weight mass*

Replaced.

- *P3, L15-16: clarify that is it 3-5 years worth of accumulation contained in the 2-m sample; specify "winter-to-winter accumulation rates"*

Both done, the paragraph now reads "Using the density data, accumulation rates at the different sites were calculated from the snow mass for the three to five years worth of accumulation contained in the top two meters of the snowpack. In the present study, we only use winter-to-winter rates (separating years at the $\delta^{18}O$ minima) – summer-to-summer values were computed as a reference but show no different behaviour."

- *P3, L21: renowned well-known*

Done.

- *P4, L23: what does the "maximal ratio of the respective accumulation rates" mean? Repective to what? Two sites next to one another?*

Yes. Clarified by adding "between two sites".

- *P6, L3: Perhaps use z_i since you are talking about depth. The x dimension (to me) indicates a direction on the surface (e.g. along your traverse).*

Changed.

- *P7, L15: Change to "The previously-calculated depth-scale density records were stacked to obtain..."*

Changed.

- *P8, L6: Increasing to 140 where?*

"at EGRIP" added for clarity.

*- P9, L1: fourth liner? Do you mean location/site?*

Yes, "liner" replaced by "site".

*- P9, L15: The statistics in this paragraph were already reported on page 7; did you intentionally do that?*

Yes. In order to improve the manuscript, one more repetition was omitted. All remaining numbers are directly refered to.

*- P9, L21: is are, summer of 2012*

Corrected.

*- P9, L27: sommer summer*

Corrected.

*- P10, L20: accustic acoustic*

Corrected.

*- Figure 8: y-axis does not have scale or units labeled*

On purpose. "Each profile is displayed at the same scale and has been centered around its mean." added to caption.

*- there are numerous instances of typos and challenging-to-read sentence structure that I have not indicated here; I recommend having a copy editor review the manuscript for those.*

*Reference: Proksch, M., Löwe, H. & Schneebeli, M. (2015). Density, specific surface area, and correlation length of snow measured by high-resolution penetrometry. Journal of Geophysical Research: Earth Surface, 120(2), 346-362.*

**Changes in manuscript**

Please find an updated version of the manuscript (changes tracked with latexdiff) attached.

**Supplement:**

[revised manuscript text omitted]